# OpenReview forum: "HieraSuite: A Holistic Toolkit for Building Versatile System-User Instruction Hierarchy"
_ICLR.cc/2026/Conference — Submitted to ICLR 2026_

### Official Review · Reviewer_7biq · 2025-10-30

**Soundness:** 2
**Presentation:** 3
**Contribution:** 1
**Rating:** 4
**Confidence:** 4

**Summary:**

This work enhances the LLMs' ability to follow the system-user instruction hierarchy by a systematic process: (1) they construct a dataset of system-user instruction pairs that include conflicting instructions, (2) they distil the GPT-4.1 model's responses for these instructions into a small 7B model, and then (3) they train a model using DPO and SFT to optimize for generations that are refined by the distilled model in the loop. Finally, experiments on a benchmark show significant improvements for different language models.

**Strengths:**

- The paper is mostly well-written with a good writing structure, minimal typos, and a documented Appendix.
- Experiments are comprehensive and cover a large space of hyperparameters and ablations.
- Curated tasks are well-aligned with the objective, and the trained model could also be helpful when released.
- The proposed approach improves the instruction hierarchy ability of language models across different settings.
- Appendix includes sufficient examples for the dataset.

**Weaknesses:**

- There is very limited originality, and the paper is just a technical report of a standard sequence of implementations over the gaps identified in Zhang et al., 2025c.
- There is a lack of transparency regarding the synthetic data creation, and it is not clear why it is needed at all, given that a lot of datasets are already available. If the contributions are mainly the new synthetic dataset, then this should be clearly written and the whole process of creation explained clearly.
- Variance in Figure 3 should be included. GPT-4.1 is used as the judge model and also the teacher model, which might lead to biased evaluations.
- The sampling strategy for HCReasoner training data is not clearly presented.
- There is a potential overlap between the HieraInstruct dataset and the benchmark datasets, which again leads to a biased evaluation. This also limits the generalization capability of the instruction hierarchy, which should also be studied in out-of-distribution tasks in addition to the generic tasks of MMLU.
- Ablation on the datasets (prompts for the preferences) used to train the HieraCRO models should be considered. Currently, it is only with HieraInstruct and HieraInstruct+HelpSteer but it is important to study what is the least amount of data (number of preferences and their category-wide diversity) that would be enough to reach a decent performance.
- The choice of using HCReasoner instead of self-improvement is not well-demonstrated in the experiments, as the performance gains are not quite enough to arguably motivate training an extra model and make the contributions even more obscure.
- Minor:
  - Figure 1 and 2 are both very hard to follow given the color choices and the density of text.

**Questions:**

- Why not just train a model with IHEval? Just full fine-tuning or preferential DPO following the strategy defined here.
- Why do we not need a clever sampling strategy (either oversampling or such) to handle conflicting pairs as opposed to non-conflicting?
- What happens if the user prompt conflicts with GPT itself? In that case, won't the distillation become a bottleneck?
- See above weaknesses

---

> ### Author Response · Authors · 2025-11-26
> **Response (1/5)**
>
> Thank you for the insightful and comprehensive feedback. **We are especially grateful that the reviewer recognizes our work as “well-written with good writing structure,” highlights our experiments and ablations as “comprehensive,” and acknowledges that our proposed approach improves the instruction-hierarchy-following ability of LLMs.** We address all questions and comments in the response below, and we are happy to clarify any additional points during the discussion period. If our responses satisfactorily address the reviewer’s concerns, we would be grateful if they would consider updating their score accordingly.
>
> ## **The Originality and Significance of Our Contributions**
>
> Thank you for raising this thoughtful point. While Zhang et al. (2025c) correctly identify the gap in IH, their work focuses exclusively on constructing an **evaluation** benchmark. We incorporate their benchmark (IHEval) as one component of our unified evaluation suite, HieraSuite, which spans ten diverse benchmarks. Crucially, Zhang et al. (2025c) do not offer any tooling or methodology for improving IH compliance in LLMs, nor do they address broader system-control dimensions such as value steerability, privacy/confidentiality, or customizable safety constraints.
>
> In contrast, our work introduces the **first holistic, principled, end-to-end framework for IH**, spanning (i) a systematically curated, novel **training** data with system-user prompt pairs, (ii) a model-agnostic optimization method for generating responses with hierarchical compliance, and (iii) a comprehensive evaluation suite that extends beyond IHEval. Together, these components make IH definable, measurable, and, **for the first time, trainable**, enabling robust and generalizable improvements in hierarchical obedience of LLMs.
>
> We therefore respectfully push back on the reviewer’s comment that “there is very limited originality, and the paper is just a technical report,” because our work provides the **first unified framework that jointly addresses IH data, training, and evaluation, an integrated solution that prior work has not offered.**
>
> *In the revised draft, we have updated both the abstract and introduction to more clearly articulate and highlight these contributions.* Below we summarize our key contributions in greater detail.
>
>
> > **We define IH as a principled, multi-faceted alignment target.**
>
> Previous work considers “instruction hierarchy” primarily in evaluation, but **not in training**, and typically focuses on relatively narrow domains such as prompt-injection defense [1] or simple system-rule following [2]. In contrast, we formalize the IH training objective as a three-part behavioral requirement: (i) overriding conflicting user instructions, (ii) integrating non-conflicting system constraints, and (iii) maintaining robustness on user-only inputs. Our data design, hierarchy reasoning, and evaluation methods directly operationalize all three. Building HieraInstruct (221K pairs), HieraConsReasoner, HieraCRO, and HieraBench required **substantial conceptual and engineering advances, resulting in the first reproducible platform for training and studying IH at scale**.
>
> > **We deliver the first empirical system showing that IH is solvable in practice.**
>
> To our knowledge, this is the **first work demonstrating that IH can be trained, improved, and evaluated consistently across model families and scales**. Our unified pipeline yields large gains, e.g., **up to +66.9% on HieraBench and +306.3% in conflict-override accuracy**, providing concrete evidence that IH is not only well-defined but also tractable. These improvements arise from coordinated innovations in hierarchical data generation, rubric-based reasoning, iterative preference construction, and unified evaluation, rather than from scale alone.
> Taken together, **these contributions constitute the first full-stack methodology for defining, training, and evaluating system–user instruction hierarchy**, going far beyond a simple reuse of existing optimization techniques.
>
> - [1] https://arxiv.org/abs/2404.13208
>
> - [2] https://arxiv.org/abs/2311.04235
>
> ## **Figure 1 and 2**
>
> Thanks for the suggestion! We’ll make sure to make Figure 1 and 2 more clear in the camera-ready with the additional one page of space we’re given.

---

> ### Author Response · Authors · 2025-11-26
> **Response (2/5)**
>
> ## **The Value and Curation Process of Synthetic Data**
>
> Thank you for emphasizing the importance of transparency in the data creation process. We fully agree with the reviewer that clarity here is essential. Recognizing this, **our paper provides substantial details in the main text (Section 2.1, L113-179) and Appendix A (page 18-29)**, but we realize that the motivation and necessity of the synthetic component of HieraInstruct may not have been sufficiently foregrounded. We are happy to clarify.
>
> > **Why synthetic data is needed.**
>
> Training LMs for robust system-level control requires alignment data that captures diverse system–user interactions. However, existing datasets contain only user instructions [1–3] or non-conflicting system add-ons [4], and none provide paired system–user instructions for broad hierarchical compliance categories. **Because such data is absent in existing resources, we constructed HieraInstruct to simultaneously cover all of the following key aspects needed for LLM training to bridge the gap**:
>
> - Conflicting system–user pairs needed to train and evaluate instruction-hierarchy override.
>
> - Aligned system–user pairs where system instructions supplement or constrain user goals.
>
> - High-privilege system rules (e.g., privacy, security, role stability, custom safety policy) paired with adversarial or circumventing user inputs.
>
> - Task-execution style system prompts where the user instruction should be treated as input content.
>
> > **How synthetic data was created, and how we ensure quality.**
>
> Section §2.1.2 and Appendix A detail our generation and filtering pipeline. Briefly:
>
> - We use a seeded, iterative generation–verification process to produce diverse system–user pairs across nine subtypes.
>
> - Synthetic data is generated only for domains or pair structures not present in existing datasets (e.g., conflicting overrides, hierarchical role constraints, cybersecurity system rules with evasive users).
>
> - For domains requiring domain expertise (e.g., cybersecurity, privacy, task execution), we apply specialized LM judges for automated validation.
>
> - All system–user pairs are expanded into four formats (aligned, conflicting, system–user combined, user-only) to enable robust IH training and generalization, when applicable.
>
> - Finally, HieraCRO creates preference pairs from synthetic data guided by hierarchical constitutions, enabling principled training rather than relying only on raw synthetic examples (Section 2.3).
>
> > **Synthetic data is not the only main contribution.**
>
> We respectfully clarify that the contribution of our work is not “mainly the new synthetic dataset.” HieraInstruct is one component of a unified, end-to-end framework (HieraSuite) that includes:
>
> - a structured multi-domain dataset (synthetic + repurposed)
>
> - a compact hierarchical constitution reasoner (HieraConsReasoner)
>
> - a principled response-optimization method (HieraCRO)
>
> - a unified 10-task evaluation suite (HieraBench)
>
> The novelty lies in **our end-to-end framework that makes IH definable, measurable, and trainable**, not in the synthetic data alone. We hope this clarifies why synthetic data is necessary and how its creation is fully documented.
>
> - [1] https://arxiv.org/abs/2411.15124
>
> - [2] https://arxiv.org/abs/2505.11475
>
> - [3] https://github.com/anthropics/hh-rlhf
>
> - [4] https://arxiv.org/abs/2405.17977
>
> ## **Sampling Strategies of Conflicting vs. Non-Conflicting Pairs**
>
> Thank you for raising this great question! In our training setup, we maintain an approximately 1:1 ratio between conflicting and non-conflicting pairs to balance three key behaviors: (1) user-only instruction following, (2) non-conflicting system–user instruction following, and (3) conflicting system–user instruction overriding. While it is important to teach the model to handle conflicts, oversampling conflicting cases distorts this balance and harms overall performance. In an ablation where we trained only on conflicting pairs, the model improved on instruction overriding (35.1% → 30.7%) but substantially degraded on non-conflicting system–user instruction following (71.9% → 76.6%). These results highlight that maintaining a balanced mix is essential for strong performance across all three dimensions.

---

> ### Author Response · Authors · 2025-11-26
> **Response (3/5)**
>
> ## **Benchmark Overlaps and Model Performance Generalizability**
>
> We appreciate the reviewer’s concern regarding potential overlap between the training data HieraInstruct and the evaluation suite, HieraSuite. We emphasize that our data-separation protocol is designed to **fully prevent leakage from training or optimization data into evaluation benchmarks**.
>
> First, **HieraInstruct (training/optimization)** and **HieraBench (evaluation)** are constructed from entirely disjoint prompt sources. HieraInstruct prompt categories are motivated by real-world LLM challenges related to system control and steerability, such as adversarial instruction following [1], privacy risks [2], cybersecurity concerns [3], and pluralistic alignment [4]. HieraBench, in contrast, is curated primarily from established external benchmarks for assessing system-level control and steerability. **No system–user instruction pair, prompt template, or underlying instance in HieraBench was used to guide the construction of HieraInstruct.** The only shared source dataset, MultifacetedBench, is partitioned into mutually exclusive splits for training and evaluation, ensuring no cross-contamination. **We additionally performed an explicit overlap analysis and confirmed that there is no verbatim overlap between HieraInstruct and any test instance in HieraBench.**
>
> Second, all synthetic data in HieraCRO is generated exclusively by applying hierarchical modifications (e.g., conflict insertion, constraint augmentation, multi-level rule perturbation) to HieraInstruct prompts only. Since HieraInstruct is fully held out from HieraBench, the synthetic generation pipeline cannot produce synthetic samples that overlap with or indirectly reconstruct evaluation items.
>
> Third, HieraBench spans broad and heterogeneous instruction categories—steerability, safety, privacy, rule-following, sourced from diverse external domains. Effective IH alignment must therefore improve performance on IH-relevant metrics while maintaining general model competence. For this reason, we also evaluate general-purpose benchmarks such as MMLU to confirm that enforcing IH compliance via HieraCRO does not harm core model capabilities. **Across the ten IH tasks in HieraSuite and 10+ general capability evaluations, our models achieve a strong Pareto frontier between IH compliance and general abilities, indicating that the improvements reflect generalizable hierarchical obedience rather than benchmark-specific tuning.**
>
> Taken together, these safeguards and comprehensive evaluations provide clear evidence that the gains do not arise from biased or overlapping benchmark construction. *We further clarify the data disjointness between HieraInstruct and HieraBench in Section 2.4 of the revised paper.*
>
> - [1] https://arxiv.org/abs/2404.13208
>
> - [2] https://arxiv.org/abs/2310.17884
>
> - [3] https://arxiv.org/abs/2306.05499
>
> - [4] https://arxiv.org/abs/2402.05070
>
>
> ## **Training Details and Performance Discussion of HCReasoner**
>
> Thank you for these thoughtful questions. We are happy to clarify both the HCReasoner data sampling procedure and the evaluation setup to address potential concerns about bias.
>
> > **Regarding data curation for HCReasoner.**
>
> Due to space constraints in the initial submission, we briefly noted in L191–194 and Appendix B.1 that HCReasoner is trained on a 100K subset randomly sampled from HieraInstruct: 23K user-only, 30K system-only, and 47K system–user hierarchy cases. To demonstrate generalizability and avoid contamination, the HCReasoner training split is strictly disjoint from the data used to train downstream LLMs (L284–285). *We have updated Section 2.2 to make the random sampling strategy and data separation more explicit.*
>
> > **Regarding the evaluation LLM judge model.**
>
> We also clarify that the evaluation of specificity, grounding, and comprehensiveness deliberately uses `gpt-5-chat-latest`  (as noted in L196 in the original paper), not GPT-4.1 as mentioned in the review. HCReasoner is therefore trained on data generated by a different model than the judge model, reducing the risk of model-specific bias in evaluation.
>
> > **Adding variances to Figure 3.**
>
> Thank you for the helpful suggestion to include variance estimates in Figure 3. *We have incorporated error bars in the revised draft.* Across all three metrics, **HCReasoner achieves not only higher average scores but also noticeably tighter error bars, reflecting more robust, reliable, and less volatile hierarchical reasoning compared to standard 0-shot baselines**. This combination of higher mean and lower variance highlights the strength of our approach: HCReasoner consistently produces high-quality, principled constitutions that generalize across diverse prompts, rather than relying on brittle or case-specific behaviors.

---

> ### Author Response · Authors · 2025-11-26
> **Response (4/5)**
>
> ## **Why Cannot We Train Models with IHEval**
>
> Thanks for suggesting this idea. **We’d like to clarify that IHEval is a benchmark constructed specifically to evaluate aspects of IH following; it is not designed or appropriate to serve as training data.** Using IHEval for fine-tuning, whether via full SFT or DPO, would directly contaminate the evaluation and invalidate the benchmark’s purpose. More importantly, IHEval is designed as a focused evaluation suite: its relatively small number of carefully curated scenarios are optimized to measure hierarchical obedience under controlled conditions. While this makes IHEval an excellent benchmark, it also means it does not provide the breadth or diversity needed to train models on the full range of system–user interactions, conflict patterns, and domain variations required for robust IH alignment.
>
> In contrast, our framework relies on HieraInstruct and HieraCRO to provide (i) large-scale, diverse system–user instruction data, (ii) contextualized hierarchical constitutions that define fine-grained behavior under conflicts, and (iii) iterative preference optimization that produces generalizable improvements. As our experiments show, these components yield consistent gains across all ten HieraBench tasks as well as 10+ general capability benchmarks, whereas training on evaluation-style data risks benchmark overfitting without improving broader system-level controllability or security.
>
> In summary, IHEval must remain strictly held out, and even aside from contamination, its limited coverage and evaluation-oriented structure make it insufficient as a standalone training signal for learning robust hierarchical obedience. Our approach is designed to avoid benchmark-specific tuning and instead deliver principled, general IH alignment.
>
> ## **Ablations of HieraCRO Models**
>
> Thank you for asking about ablations of HieraCRO Models. We would like to emphasize that our paper already includes **40+ extensive and multi-axis ablations** on the data used to train HieraCRO, far beyond the two mixtures highlighted in the review.
>
> Specifically, we evaluate how **different data sources, domains, and generation strategies affect IH alignment**, including:
>
> - Constitution guidance ablations (No-Cons., GPT-Cons., HCReasoner-Cons.) in Table 3, showing that constitution quality, not just data volume, is a critical factor.
>
> - Iterative vs. single-pass data generation (No-Iter.), also in Table 3, demonstrating the impact of iterative refinement on the resulting preference data.
>
> - Self-Improvement vs. HCReasoner-Guided data creation in Figure 5 and Tables 24–26, showing clear, consistent gaps in favor of guided, structured preference generation.
>
> - Domain-specific ablations across the four components of HieraInstruct, including System Constraints, Privacy/Security, Steerability, Task Execution (Table 3), revealing that single-domain mixtures lead to unbalanced performance, whereas full-domain mixtures provide robust IH adherence.
>
> - Mixture composition and recipe ablations across {IH vs. HelpSteer3 vs. IH+HS} × {DPO vs. SFT} × {LoRA vs. Full FT} (Figure 4, Tables 15–17), illustrating how data composition interacts with training strategy and affects both IH metrics and general capabilities.
>
> Together, these experiments cover **40+ configurations** and jointly probe the effects of data domain, structure, source, guidance paradigm, and training setup. They provide strong evidence that the structure, data categories, and hierarchical specificity of IH-relevant data are what drive HieraCRO’s gains.
>
> While we agree that examining minimal data requirements, the scaling of category-level diversity, and broader ablations across additional model families would yield valuable insights, we leave these empirical extensions to future work. In our existing experiments, models are trained on $\sim$40K DPO pairs, and leveraging all categories in HieraInstruct leads to balanced and consistently improved performance. **Importantly, HieraInstruct, HCReasoner, and HieraCRO already provide the complete set of ingredients needed for future research and production systems to perform such ablations and uncover new insights.**

---

> ### Author Response · Authors · 2025-11-26
> **Response (5/5)**
>
> ## **Justification of Why We Need An External HCReasoner Model**
>
> We thank the reviewer for raising this point. We agree that any additional component, such as HCReasoner, should be justified by clear empirical benefits. Our experiments were designed precisely to test this, and the results show that HCReasoner plays a formative role in improving instruction-hierarchy adherence in ways that self-improvement does not match.
>
> First, **self-improvement alone provides gains, but they are consistently smaller and less stable**. As shown in Figure 5 and Tables 24–26 (Appendix F), across Qwen-2.5 7B/14B/32B, self-improvement yields 9.5–13.9% average relative improvement across all ten tasks in HieraBench. We respectfully disagree with the comment that these improvements are insignificant; they are both meaningful and consistent.
>
> Second, **HCReasoner generates qualitatively superior constitutions**, which directly enable higher-quality preference data. As shown in Figure 3, HCReasoner-7B outperforms vanilla Qwen-7B without training by a large margin across specificity (1.80 to 1.93), compositionality (1.72 to 1.85), and grounding (1.76 to 1.95). These improvements translate directly into downstream gains. In Table 3, removing constitutions (“No Cons.”) leads to substantial performance drops, for example from 67.0 to 61.3 on IHEval and from 0.74 to 0.66 on PurpleLlama. This ablation makes clear that HCReasoner is a primary driver of the observed robustness improvements.
>
> Third, **self-improvement is fundamentally limited by the base model’s own weaknesses**, because it relies on the same model as reasoner, reviser, and verifier. This creates a feedback loop in which the model must detect and correct failure modes that it does not reliably recognize. HCReasoner provides an externalized and high-specificity hierarchical rubric that is out of distribution for the base model, allowing the system to make corrections that it cannot produce through self-improvement alone.
>
> Fourth, the reviewer expresses concern about training an additional model. We emphasize that **HCReasoner is intentionally compact (7B or 14B), trained once, and then reused across all models and experiments. We also release this model publicly.** The overhead is minimal, yet the gains are large and consistent across model families (Llama, Qwen, Mistral) and scales (7B to 32B). HCReasoner also serves multiple purposes: in addition to improving preference data for HieraCRO, it produces high-fidelity hierarchical constitutions that are valuable for interpretability and diagnostics.
>
> Finally, we want to highlight that HCReasoner provides a reusable mechanism for principled content specification, conflict detection, and hierarchical interpretation across a wide range of scenarios, including privacy, safety, pluralistic value steering, and system-constraint enforcement. In this sense, **HCReasoner is not simply an auxiliary component for data creation. It is a research artifact with value that extends well beyond its role within HieraCRO.** For example, the itemized constitutions produced by HCReasoner can be directly repurposed to construct interpretable, rubric-based evaluation metrics, similar in spirit to rubric-anchored evaluations in RLVR or checklist-style evaluation frameworks. This enables transparent and fine-grained measurement of model behavior, rather than relying solely on coarse binary pass or fail outcomes.
>
> In summary, although self-improvement contributes some gains, HCReasoner is essential for achieving the strongest, most reliable, and most generalizable improvements in instruction-hierarchy compliance. It substantially outperforms self-improvement, and its inclusion is therefore both empirically justified and central to the methodological contribution of the paper.
>
>
>
> ## **What If User Prompt Conflicts with GPT**
>
> Thanks for this thoughtful question. We interpret the concern as asking whether the teacher model’s own stylistic biases or preferences might conflict with user instructions and thus create a bottleneck during distillation. We would like to clarify that GPT is used only to generate or augment the prompt pairs (system–user inputs), not the responses that our models are trained to imitate. All training responses are produced through the HieraCRO pipeline, which is guided by HCReasoner’s itemized, hierarchy-aware constitutions. HieraCRO performs iterative revision and verification using the base model being improved (e.g., off-the-shelf Qwen), ensuring that the resulting preference pairs reflect principled system–user hierarchy rules rather than GPT-specific response tendencies. This design explicitly avoids the kind of “teacher-style bottleneck” the reviewer is concerned about. Empirically, we observe no such bottleneck: HieraCRO consistently improves conflict resolution, aligned system-following, and robustness across all model families (Table 2).

---

### Official Review · Reviewer_JWgZ · 2025-10-31

**Soundness:** 3
**Presentation:** 3
**Contribution:** 2
**Rating:** 6
**Confidence:** 3

**Summary:**

This paper proposes a framework to evaluate and improve the LM's ability to follow the instruction hierarchy. The author first created HieraInstruct to collect system-user instruction pairs, and use them to train response reasoners to help improve the instruction hierarchy following ability of the model. The improvement is done by iterative refinement of the model response and using contrastive pairs to finetune the model with DPO. The improvement was shown on diverse instruction following datasets.

**Strengths:**

The paper did a comprehensive analysis and data collection to help improve the instruction hierarchy following ability.
The data collection serves a good evaluation and training framework for future LLM design.
The ablation study could support the experiment design well.

**Weaknesses:**

The main weakness is the over-reliance on the LLM evaluation over the whole pipeline. This is understandable as large-scale data collection requires a scalable approach for evaluation. However, it would be great if there is any human evaluation involved, or at least more focus on the LLM-as-a-Judge analysis.

Currently, the evaluation part has little information on how the evaluation was done.

**Questions:**

see weakness above

---

> ### Author Response · Authors · 2025-11-26
> **Response (1/2)**
>
> Thank you for the positive and constructive feedback. We are especially grateful that the reviewer recognizes our work as providing **“comprehensive analysis and data collection,”** offering **“a good evaluation and training framework for future LLM design,”** and **supporting the experiment design through detailed ablations**. We address all questions and comments thoroughly in the response below, and we are happy to clarify any additional points during the discussion period. If our responses satisfactorily address the reviewer’s concerns, we would be grateful if they would consider updating their score accordingly.
>
> ## **Clarification on Evaluation Quality**
>
> ​​Thank you for highlighting this important point regarding evaluation and the role of LLM judges. We fully agree that large-scale IH research must carefully justify how model-based evaluation is used. **To clarify: in our pipeline, LLM judges are used only in controlled and localized components, not as the primary mechanism for evaluating re-aligned LLMs.** Specifically, judge models are applied only (i) to evaluate HCReasoner quality during its development, and (ii) to filter two narrow subcategories of HieraInstruct (Permissible Use Cases and Cybersecurity) where domain-specific constraints require automated screening. All final model evaluation is conducted using HieraBench, a 10-task unified suite combining extensive programmatic and rule-based checks (e.g., VerSR, RuLES, PurpleLlama) and structured metrics inherited from prior benchmarks (e.g., IHEval, SysBench), with LLM-as-a-judge used strictly only where required by the benchmark’s original design. **This multi-source evaluation substantially reduces dependence on any single judge model.**
>
> ### **New Human Evaluation of HieraInstruct**
>
> To further validate data quality, we conducted a **new human evaluation** of system–user instruction pairs in HieraInstruct. We sampled 900 data points (100 from each subcategory) and recruited qualified annotators on Prolific who met strict criteria: **English as a first language, ≥ high-school education, approval rate > 99%, and ≥ 1000 completed tasks.**
>
> Each system–user pair was evaluated independently by **three annotators**, who rated:
>
> - quality of the system prompt
> - quality of the user prompt
> - quality of the combined pair
>
> We take the **majority vote** as the final label. Annotators judged whether each item is **clear, coherent, meaningful, and appropriate for use in language-model training**. Results show:
>
> - **97.7%** of System Prompts
> - **91.7%** of User Prompts
> - **90.1%** of Combined Instructions
>
> meet all quality criteria. **This directly confirms the high quality of HieraInstruct.**

---

> > ### Author Response · Authors · 2025-11-29
> > **Response (2/2)**
> >
> > ### **Details on Each Use of LLM Judges**
> >
> > > **Use of LLM Judges for HCReasoner Evaluation**
> >
> > To justify the need for a trained hierarchical reasoner, we used `gpt-5-chat-latest` to compare zero-shot LMs vs. trained HCReasoner on three rubric dimensions—specificity, grounding, and comprehensiveness. This judge usage is limited to comparing relative quality of the reasoners, not evaluating final aligned LLMs. The goal is simply to demonstrate that HCReasoner meaningfully improves the quality and structure of constitutions (§2.2, Figure 3 in the paper.
> >
> > > **LLM Judges for Filtering Two Specialized Subcategories of System–User Pairs**
> >
> > During synthetic system-user prompt collection phase, we apply LLM judges to filter two subcategories of complex data types, inducing Permissible Use Cases (LLM Judge Prompt in Figure 23) and Cybersecurity (LLM Judge Prompt in Figure 26). Specifically:
> >
> > (1) Permissible Use Cases
> >
> > During the development of this data category, we observed that some synthetic system instructions implicitly required capabilities beyond standard text-only LLM usage—such as multimodal processing, agentic memory, or retrieval mechanisms. Because our goal is to construct data suitable for general-purpose LLMs, we filter out any cases that fall outside pure language-based capabilities. Accordingly, we apply an LLM-judge filtering step, guided by precise criteria (e.g., all tasks in the system prompt must be achievable solely through language understanding, reasoning, and generation) and illustrated through explicit examples:
> >
> > - Performing real-world physical actions
> > - Operating hardware or devices
> > - Executing code in a live environment
> > - Making financial transactions
> > - Browsing the internet in real time
> > - Accessing private or external databases not provided in context
> >
> > (2) Cybersecurity
> >
> > For the cybersecurity subcategory, where concepts are more complex and domain-specific, we apply an additional verification stage using carefully prompted LLM judges. These judges evaluate each candidate along five dimensions: cybersecurity relevance, risk specificity, concreteness, realism, and meaningfulness. To maintain data quality, we provide precise definitions for each dimension and **retain only data points that satisfy all criteria, avoiding noise that might arise from partial passes.**
> > Because both filtering stages are supported by thorough prompt engineering and use the strong GPT-4.1 model as the judge, this process substantially improves the overall quality of the resulting system–user prompt subset.
> >
> > > **LLM judges are used in HieraBench only when inherited from the original benchmark definitions, in order to preserve their standardized evaluation protocols.**
> >
> > To clarify the role of LLM judges in our evaluation, **HieraBench employs LLM-as-a-judge strictly in the tasks where such judging is already part of the benchmark’s original design**, and we do not introduce any new LLM-judged components beyond what those benchmarks specify. Several existing benchmarks integrated into HieraBench, such as RoleMRC, and Multifaceted-Bench, and CoSA, rely on LLM evaluators because their underlying tasks (e.g., role consistency, persona steering, value-shift assessment) do not admit deterministic or programmatic scoring. To preserve standardized evaluation protocols, ensure comparability with prior work, and avoid inadvertently biasing results by modifying evaluation criteria, we faithfully adopt their original judge-based components without alteration. Importantly, a subset of HieraBench tasks, such as VerSR and PromptSteering, relies entirely on rule-based, programmatic, or heuristic scoring, not LLMs. In summary, HieraSuite uses LLM judges only when mandated by inherited benchmarks to maintain methodological consistency and rigor.

---

### Official Review · Reviewer_9RR4 · 2025-10-31

**Soundness:** 3
**Presentation:** 3
**Contribution:** 2
**Rating:** 4
**Confidence:** 3

**Summary:**

The paper presents a large-scale framework for constructing and evaluating instruction hierarchices in LLMs. The is a very important question that poses safety concern, so the authors propose HieraSuite, a holistic toolkit consisting of dataset, reasoning model, an iterative response optimization framework and a suite of evaluation benchmarks and demonstrated that the proposed model achieved larged gains on their benchmark while maintaining general performance.

**Strengths:**

1. Quality: The paper is technically sound, with very extensive and comprehensive empirical experiments and a consistent methodology across diverse model backbones. Finishing this project requires lots of computing resources.
2. Clarity: the paper is well-written and visually organized. Although I feel Figure 1 is a bit too dense but doesn’t hurt the overall readability. Details are provided so that I believe the results are reproducible.
3. Significance: given that IH is a very important topic to improve LLM safety, this paper contributes a very valuable infrastructure for future research.

**Weaknesses:**

1. Originality: the contribution is primarily empirical and integrative, not methodological contributions. For example, it reuses existing frameworks including DPO-style preference optimization, constitution guided scoring [1], iterative refinement [2] with minimal algorithmic innovation. Besides, for the HierBench, most of the benchmarks are reused public benchmarks.
2. I’m worried that much of the reported gain seems to come from scale and data curation, rather than a new training principle.
3. It would be helpful if authors could clarify how evaluation benchmark datasets are selected and to what extent they overlap with the training or optimization data used in HierCRO (especially since you feed synthetic data which are harder to control). Given that both the method and the benchmarks emphasize hierarchical obedience, it will be nice to provide evidence or analysis demonstrating that the reported improvements are not primarily due to benchmark-specific tuning.

[1] Bai, Yuntao, et al. "Constitutional ai: Harmlessness from ai feedback." arXiv preprint arXiv:2212.08073 (2022).

[2] Madaan, Aman, et al. "Self-refine: Iterative refinement with self-feedback." Advances in Neural Information Processing Systems 36 (2023): 46534-46594.

**Questions:**

In line 175, when authors mentioned: “For practical use, LMs must (i) override conflicting user instructions, (ii) integrate supplementary non-conflicting system constraints, and (iii) perform robustly on user-only inputs.” I wonder if the model were trained in this way, how does the model perform on user system conflict detection, and people could prefer an LLM that explictly acknowledge the conflict first without continuing directly to override user instructions. Otherwise, you could imagine a user will get confused because they feel the model is not following provided instructions.

---

> ### Author Response · Authors · 2025-11-26
> **Response (1/3)**
>
> Thank you for the insightful and thorough feedback! We appreciate the reviewer’s recognition of our work as **“a very valuable infrastructure for future research”** on IH, which is **“a very important topic to improve LLM safety.”** We are also grateful for the positive remarks that the paper is **“technically sound,”** supported by **“extensive and comprehensive”** experiments, and **“well-written and visually organized.”** We address all questions and comments in detail below, and are happy to clarify any further points during the discussion period. If our responses adequately resolve the concerns, we would be grateful if the reviewer would consider updating their score.
>
> ## **The Originality and Significance of Our Contributions**
>
> Thank you for raising this thoughtful point! While HieraCRO builds on established alignment components, the core novelty and scientific contribution of our work is **providing the first holistic, principled, end-to-end framework that makes IH a definable, measurable, and trainable alignment objective**. To our knowledge, **no prior work has addressed IH in a systematic or unified way**. *In the revised draft, we have updated both the abstract and introduction to more clearly articulate and highlight these contributions.* Below we summarize our key contributions in greater detail.
>
> > **(1) We define IH as a principled, multi-faceted alignment target.**
>
> Previous work considers “instruction hierarchy” primarily in evaluation, but **not in training**, and typically focuses on relatively narrow domains such as prompt-injection defense [1] or simple system-rule following [2]. In contrast, we formalize the IH training objective as a three-part behavioral requirement: (i) overriding conflicting user instructions, (ii) integrating non-conflicting system constraints, and (iii) maintaining robustness on user-only inputs. Our data design, hierarchy reasoning, and evaluation methods directly operationalize all three. Building HieraInstruct (221K pairs), HieraConsReasoner, HieraCRO, and HieraBench required **substantial conceptual and engineering advances, resulting in the first reproducible platform for training and studying IH at scale**.
>
> > **(2) We adapt existing techniques to a new hierarchical alignment objective.**
>
> Although HieraCRO builds on components from existing alignment paradigms, its novelty lies in repurposing these methods to address hierarchical, context-dependent conflicts, **where standard alignment data and objectives are ineffective**. Central to this is HieraConsReasoner, which produces contextualized, itemized hierarchical constitutions that serve as fine-grained rubrics for interpreting system–user relationships. This enables precise conflict resolution that existing alignment frameworks cannot support. In essence, **HieraCRO turns general alignment tools into a domain-specific optimization pipeline tailored for IH**.
>
> > **(3) We deliver the first empirical system showing that IH is solvable in practice.**
>
> To our knowledge, this is the **first work demonstrating that IH can be trained, improved, and evaluated consistently across model families and scales**. Our unified pipeline yields large gains, e.g., **up to +66.9% on HieraBench and +306.3% in conflict-override accuracy**, providing concrete evidence that IH is not only well-defined but also tractable. These improvements arise from coordinated innovations in hierarchical data generation, rubric-based reasoning, iterative preference construction, and unified evaluation, rather than from scale alone.
>
> Taken together, **these contributions constitute the first full-stack methodology for defining, training, and evaluating system–user instruction hierarchy**, going far beyond a simple reuse of existing optimization techniques.
>
> > **Regarding reusing existing benchmarks.**
>
> Finally, we would like to clarify that while nine of the ten underlying tasks originate from existing resources, **this is the first work to curate, modify, and organize them specifically for evaluating the multi-faceted components of IH**. *Unified benchmark suites constructed from established resources are both standard and often transformative in alignment and evaluation research*, examples include DecodingTrust [3] (NeurIPS 2023 Outstanding Paper), BIG-BENCH [4], and HELM [5], all of which synthesize and adapt existing benchmarks into coherent evaluation platforms that enable systematic analysis of model limitations. For these reasons, we respectfully disagree that the reuse and adaptation of existing benchmarks should be viewed as a limitation; rather, it is a strength that enables reproducible, community-grounded, and comprehensive evaluation of IH for the first time.
>
> - [1] https://arxiv.org/abs/2404.13208
>
> - [2] https://arxiv.org/abs/2311.04235
>
> - [3] https://arxiv.org/abs/2306.11698
>
> - [4] https://arxiv.org/abs/2206.04615
>
> - [5] https://arxiv.org/abs/2211.09110

---

> ### Author Response · Authors · 2025-11-26
> **Response (2/3)**
>
> ## **Benchmark Overlaps and Model Performance Generalizability**
>
> We appreciate the reviewer’s concern regarding potential benchmark-specific tuning, especially given that HieraCRO employs synthetic preference data. We emphasize that our data-separation protocol is designed to **fully prevent leakage from training or optimization data into evaluation benchmarks**.
>
> First, **HieraInstruct (training/optimization)** and **HieraBench (evaluation)** are constructed from entirely disjoint prompt sources. HieraInstruct prompt categories are motivated by real-world LLM challenges related to system control and steerability, such as adversarial instruction following [1], privacy risks [2], cybersecurity concerns [3], and pluralistic alignment [4]. HieraBench, in contrast, is curated primarily from established external benchmarks for assessing system-level control and steerability. **No system–user instruction pair, prompt template, or underlying instance in HieraBench was used to guide the construction of HieraInstruct.** The only shared source dataset, MultifacetedBench, is partitioned into mutually exclusive splits for training and evaluation, ensuring no cross-contamination. **We additionally performed an explicit overlap analysis and confirmed that there is no verbatim overlap between HieraInstruct and any test instance in HieraBench.**
>
> Second, all synthetic data in HieraCRO is generated exclusively by applying hierarchical modifications (e.g., conflict insertion, constraint augmentation, multi-level rule perturbation) to HieraInstruct prompts only. Since HieraInstruct is fully held out from HieraBench, the synthetic generation pipeline cannot produce synthetic samples that overlap with or indirectly reconstruct evaluation items.
>
> Finally, the instruction categories in HieraBench (e.g., steerability, safety, privacy, rule-following) are intentionally broad, heterogeneous, and sourced from diverse domains. HieraCRO’s strong performance on general-purpose evaluations further indicates that the method improves general hierarchical obedience, rather than exploiting benchmark-specific patterns.
>
> Taken together, these safeguards provide clear evidence that the observed gains do not arise from benchmark-specific tuning. *We further clarify the data disjointness between HieraInstruct and HieraBench in Section 2.4 of the revised paper.*
>
> - [1] https://arxiv.org/abs/2404.13208
>
> - [2] https://arxiv.org/abs/2310.17884
>
> - [3] https://arxiv.org/abs/2306.05499
>
> - [4] https://arxiv.org/abs/2402.05070
>
> ## **Clarification Around Reported Gains Come from Scale and Data Curation, Rather Than New Training Algorithm**
>
> Thank you for raising this thoughtful concern. We would like to emphasize that IH is fundamentally unlearnable without structured hierarchical training data, yet **no existing alignment training corpus includes the essential ingredients for IH**: current datasets contain only user instructions [1–3] or non-conflicting system add-ons [4], and none include paired system–user instructions covering broad hierarchical compliance categories. **Our work introduces the first principled framework for generating conflict-aware preference data, producing the only publicly available training dataset for IH.** We show the effectiveness of our training pipeline beyond simple scaling of data: in Table 3, when data volume is held constant, removing hierarchical constitutions or iterative refinement leads to steep performance collapses, directly showing that the gains arise from the algorithmic structure of our data generation, not from scale.
>
> More importantly, this reflects a well-established pattern in modern alignment research: breakthroughs such as RLHF, RLAIF, Constitutional AI, Self-Refinement, and GRPO/RLVR did not arise from inventing new primitive loss functions, but from designing principled algorithms that curate, refine, or generate higher-quality preference signals, whether online or offline. The field widely recognizes that when no training signal exists for a capability, the core scientific contribution lies in creating the algorithm that produces that signal. IH is precisely such a case: there is currently no pre-existing supervision for hierarchical conflicts, system–user override, or multi-layered instruction dynamics. In this context, our hierarchical-constitution generation coupled with iterative conflict-aware preference construction is not “data curation”; it is the central training principle that makes IH learnable at all. The fact that this approach yields consistent and substantial gains is therefore not incidental; it provides strong empirical evidence that we have introduced the correct IH-specific training mechanism, offering both the scientific foundation and the practical pathway for building Instruction Hierarchy into LLMs.
>
> - [1] https://arxiv.org/abs/2411.15124
>
> - [2] https://arxiv.org/abs/2505.11475
>
> - [3] https://github.com/anthropics/hh-rlhf
>
> - [4] https://arxiv.org/abs/2405.17977

---

> > ### Author Response · Authors · 2025-11-26
> > **Response (3/3)**
> >
> > ## **Clarification of Our Response UX Design**
> >
> > Thank you for this insightful question; it highlights a crucial dimension of IH for real-world deployment and user experience. We fully agree that in many practical applications, users can benefit when an LLM explicitly acknowledges a system–user conflict (for example, “I see you asked me to X, but my system-level instruction requires Y, so I must follow Y for safety”). Such transparency can meaningfully strengthen clarity and user trust.
> >
> > In this work, however, our **primary technical objective was hierarchical compliance, which ensures that the model reliably overrides conflicting user instructions**. This capability is the necessary prerequisite for any safe and steerable IH behavior. Accordingly, our experiments evaluate the model’s terminal behavior, meaning whether it follows the higher-level instruction. Explicit conflict acknowledgment is an orthogonal capability that can be layered on top of compliance rather than replacing it.
> >
> > That said, we strongly agree that explicit conflict detection and communication is valuable, and represents an exciting direction for future development. Our framework is naturally extensible to support this:
> >
> > - **Rubric Extension**: The constitutional rubrics in HieraConsReasoner can be augmented with a criterion that requires the model to explicitly identify and acknowledge system–user conflicts before producing the compliant output.
> >
> > - **Fine-grained Conflict Analysis**: Itemized constitutions already generated by HCReasoner can directly serve as ingredients for a user-facing explanation. For example, the rubric derived from the User Instruction Only perspective can be used as X and the rubric derived from the System Instruction Only perspective can be used as Y in the templated clarification message.
> >
> > - **Refinement Integration**: This new criterion can be incorporated into the iterative refinement loop in HieraCRO, enabling the model to learn transparency and compliance at the same time.
> >
> > *We have added this important and promising avenue for future work to the discussion section in Appendix A.*

---

### Meta-Review · Area_Chair_kjio · 2026-01-03

**Summary:**

This paper provides a comprehensive framework for constructing and evaluating the Instruction Hierarchy (IH), covering data curation, reasoning models, optimization, and a unified benchmark. The reviewers raised many useful questions. I’ve chosen a few that I think the authors could have addressed better.

(1) The methodological contributions are limited. The training pipelines and the methodological framing seem to pre-exist. This is raised by both reviewers 9RR4 and 7biq.

(2) It is not clear why synthetic data creation is needed or whether it is the core contribution in the dataset curation process, given the many pre-existing datasets.

(3) Potential method-benchmark overlap and benchmark-specific tuning. This was raised by both reviewers 9RR4 and 7biq.

(4) Ablation on the datasets (prompts for the preferences) used to train the HieraCRO models.

**Reviewer Concerns:**

The authors provided the following answers.

(1) The authors claimed that the paper’s main contribution is not methodological but rather elevating IH from an evaluation concept to a trainable alignment objective. It also shows that IH is “solvable” in the sense that it can be trained and improved. While this is a convincing way to reframe novelty, the key question remains: whether the introduced framework is a fundamentally new alignment method or a carefully engineered composition of known ones. For example, could simpler pipelines achieve most of the gains?

(2) The authors justify the need for synthetic data but do not empirically test alternatives or demonstrate how robust or indispensable their particular synthetic data generation procedure is.

(3) The authors provided details on dataset curation, aiming to show that the observed gains are not due to dataset-specific tuning. While I can understand the authors’ intent, I feel the reviewer’s question is more profound: is the evaluated objective a rather limited concept of “hierarchical obedience” rather than genuine generalization, such as robust instruction-following?

(4) While the authors pointed out that they do have quite extensive ablations on the data used to train the HieraCRO models, I feel the reviewer 7biq is asking specifically, “what is the least amount of data” that would be enough. I believe the authors understand that they haven’t provided the exact answer.

**Reviewer Scores:**

Given the discussion above, my impression is that the two reviewers who assigned scores of 4 are unlikely to substantially increase their ratings, even after engaging with the authors. I therefore expect the review scores to remain largely unchanged. More importantly, the paper may have been submitted to a less suitable track. The work appears to be mainly a large-scale dataset and benchmark contribution, while the theoretical and methodological innovations are relatively limited.

---

### Decision · Program_Chairs · 2026-01-26

Reject